# “Have You Seen This Drivel?” A Comparison of How Common Health Issues Are Discussed within Brachycephalic and Non-Brachycephalic Dog Breed Groups on Social Media

**DOI:** 10.3390/ani14050757

**Published:** 2024-02-28

**Authors:** Kitty Phillips, Carrie Stewart, Taryn Johnston, Daniel S. Mills

**Affiliations:** 1Animal Behavior, Cognition & Welfare Group, University of Lincoln, Lincoln LN6 7DL, UK; kittyphillips95@gmail.com (K.P.); dmills@lincoln.ac.uk (D.S.M.); 2Department of Marketing, Languages and Tourism, University of Lincoln, Lincoln LN6 7DL, UK; tjohnston@lincoln.ac.uk

**Keywords:** animal welfare, brachycephalic, pet ownership, owner perception, health, health information, social support, social media

## Abstract

**Simple Summary:**

Breeds of dog with short noses, such as the French Bulldog, are increasingly popular. However, a number of health problems resulting from having this facial structure are leading to growing concerns about the welfare of these breeds. Understanding the owners of these breeds can provide important information as to how best to inform owners of these risks. In this study we wanted to know if owners of short-nosed dog breeds may use dog breed groups on social media in a different way from owners of non-short-nosed dog breeds (e.g., Labradors). We selected six dog breeds (three short-nosed and three non-short-nosed) and identified two breed groups for each breed. We then extracted the first 20 posts in relation to common health issues affecting these breeds. Owners of non-short-nosed faced dogs appeared to know more about common health issues affecting their breed than owners of short-nosed breeds. Owners of short-nosed dog breeds elicit more social support from their social media breed group, than the owners of non-short-nosed dog breeds. There appears to be greater emotionality of content associated with ownership of a short-nosed breed.

**Abstract:**

As brachycephalic dog breed ownership increases, there is a growing concern for the welfare of these breeds due to extreme brachycephalism. Understanding the motivations and behaviours of those choosing to own these breeds is important if we wish to address these concerns. The aim of this study was to investigate how owners of brachycephalic and non-brachycephalic dog breeds use social media dog breed groups to discuss common health issues. The purpose of Facebook posts in relation to common health issues, owner awareness of health issues and the role of Facebook facilitated social support were explored between brachycephalic and non-brachycephalic dog owners. Twelve Facebook breed goups were selected (brachycephalic breed groups, *n* = 6, non-brachycephalic breed groups, *n* = 6). Using key word searches we extracted the first twenty posts from each group. Thematic analysis revealed three overarching themes: advice seeking, advice giving and community bonding mechanisms. Whilst the purpose of posting did not differ between groups, non-brachycephalic owners appeared to display greater awareness of breed-specific health issues, whilst social support played a more prominent role in brachycephalic groups. This research highlights that social media groups can act as platforms for knowledge exchange and emotional support. These could be utilised by owners, veterinarians and welfare organisations to more effectively enhance dog health and wellbeing.

## 1. Introduction

There are approximately 11 million pet dogs in the UK, with pet dog ownership increasing during the COVID-19 pandemic [1]. Trends in breed popularity have changed; brachycephalic dog breeds (dogs with shortened muzzles) such as English Bulldogs, French Bulldogs and Pugs have increased by approximately 180% over the last ten years in the UK, overtaking long standing popular breeds such as Labrador Retrievers [2,3]. However, extreme brachycephalism has been associated with a number of health issues, not least airflow resistance, obstructed breathing, secondary gastrointestinal disorders and hiatal hernia, commonly referred to as brachycephalic obstructive airway syndrome (BOAS) [4,5,6]. Severe cases may lead to upper airway obstruction and can be fatal [4,5,6] with the average life expectancy of a French Bulldog now being four years [7]. The increasing severity of these health problems within these breeds has led to some veterinarians and welfare organisations stating that the health of these breeds is “too compromised to continue breeding” [8]. Despite welfare campaigns attempting to educate owners about these health issues [9,10,11,12], the popularity of these breeds continues to increase [10,13]. On the face of it, for owners of these breeds, the health and longevity of the breed seems to be secondary to their emotional attraction and aesthetic appeal [3,13].

To understand how owners process information relating to health risks in their chosen breed, we need to understand how pet dog owners access and use health-related information. Pet owners have shown a preference for information which is more accessible, faster and anonymous [14]. Many owners are turning to online sources to gain more information about their pet. One such source is social media platforms which provide pet owners with new ways to connect with each other. Social media groups can facilitate a virtual community, where users can make meaningful social connections and gain peer support [15] providing enhanced learning opportunities and access to health-related information [16,17]. Social media groups offer broader perspectives, anecdotal evidence and quick information gathering, potentially enhancing decision-making and altering attitudes [18]. It has been found that pet owners consider social media as a primary source of health information [14,19,20,21,22,23,24,25,26]. Paradoxically, many owners do not see health information on social media as reliable [19] and have trouble understanding the information [24,27]. Somes owners may change their health care decisions, not always positively, due to the desire to conform and be accepted by a group [28]. Social media can be a powerful platform for raising awareness about companion animal health and welfare. It is the fastest and most efficient way of communicating with the public and can enable change in behaviour patterns and opinions [29]. Understanding the mechanics of pet health information-seeking via social media can help veterinarians and animal welfare organisations tailor their communication strategies to ensure owners have access to accurate information and guidance.

While evidence is emerging around how and why pet owners use social media for pet health information, there is limited research exploring differences between pet owners. This is an important gap in the literature that warrants further attention. Pet owners are a diverse population; we know that motivations to purchase individual dog breeds vary greatly [13]. Concern over qualities such as health-risks are not given similar weight between pet dog owners of differing breeds [13]. Therefore, it is seems reasonable to enquire whether owners of certain dog breeds utilise pet health information on social media differently from other breeds The aim of this qualitative content analysis study was to explore how brachycephalic and non-brachycephalic dog breed owners discuss common health issues in social media groups, through the following objectives:Compare the purpose of Facebook (FB) posts in relation to common health issues between brachycephalic and non-brachycephalic dog owners in breed-specific groups.Explore owner awareness of health issues affecting their breed and identify if this varies between brachycephalic and non-brachycephalic dog owners.Explore the role of social support sought and provided by dog owners within FB breed-specific groups and identify if this varies between brachycephalic and non-brachycephalic dog owners.

Exploring this may reveal important insights into the information and other needs of different breed ownership groups, their knowledge of pet health issues, their reliance and trust in social media-provided health information, which may be useful in the design of future health and welfare interventions.

## 2. Materials and Methods

### 2.1. Design

An exploratory qualitative content analysis with data extracted from Facebook social media groups using key word searches.

### 2.2. Data Collection

#### 2.2.1. Selecting the Breed and Health Conditions

The dog breeds and health issues to be investigated were determined through a collaborative decision-making process within the research team, which included a clinical veterinary academic. The condition of hip dysplasia was chosen for the non-brachycephalic groups, which was to include breed groups for Golden Retriever, Labrador and German Shepherd. The condition of brachycephalic obstructive airway syndrome (BOAS) chosen for the brachycephalic groups which was to include English Bulldog, French Bulldog and Pug. The breeds were selected based on their popularity and the conditions selected to represent prevalent conditions amongst the respective group of breeds.

#### 2.2.2. Selecting the Dog Breed Facebook Groups

A search on Facebook for dog breed groups for each respective breed was conducted. For each of the six selected breeds, two Facebook groups which appeared primarily UK-based were joined. These groups were the first two that appeared when breed groups were searched for on FB to avoid introducing biases. The primary researcher (KP) joined twelve Facebook (FB) groups.

#### 2.2.3. Selecting the Key Words

Key words to identify the first 20 relevant posts in each group were determined through a collaborative decision-making process within the research team, which included experts in veterinary medicine, qualitative research, marketing and social media research. Key words were selected to include medically correct terminology and lay terminology known to be used by owners when discussing these health conditions. Sense checking in the Facebook groups was conducted to ensure they were appropriate and providing relevant results. For the brachycephalic breeds, we used Brachycephalic Obstructive Airway Syndrome OR BOAS OR Nose job OR Rhinoplasty OR Staphylectomy OR Soft palate OR Breathing OR issues OR surgery OR Nostrils OR nares OR airways. For the non-brachycephalic breeds, we used Hip dysplasia OR Hip replacement OR Hip operation OR Bad hips.

#### 2.2.4. Data Extraction

Each included Facebook group was searched using the keywords described above. The first twenty posts, per keyword, within each group that met the inclusion criteria were copied and pasted into a Microsoft Excel (version 2312) file. To be included, posts had to contain the predetermined keyword and be posted between 2020 and 2023. As we also wanted to explore the interactive nature of the group, posts with less than five comments were excluded. Following analysis, we were confident data saturation had occurred and there was no benefit to collecting further posts.

#### 2.2.5. Ethics

This study has been reviewed and approved by the University of Lincoln Ethics Committee (UOL2023_1069). As the social media groups and group members posting comments used in this analysis did not provide informed consent direct quotations have not been published in this manuscript.

### 2.3. Data Analysis

The collected posts were analysed using qualitative thematic analysis. Quantitative analysis was limited to basic descriptive analysis to describe the most common themes by reporting their frequency and comparing frequencies between Facebook groups and between brachycephalic and non-brachycephalic groups. Thematic content analysis was used to qualitatively analyse the content of the posts. The 6-Phase Braun and Clarke [30] approach to thematic analysis was adopted, described in Figure 1. This approach involves inductive (themes are developed based upon similar issues or points within the data) and deductive methods (applying existing concepts and ideas from previous work) for developing themes, useful units of data to be used to demonstrate and explain the findings. KP and CS reviewed the data (comments extracted from the FB groups) and noted the points being made in each post. KP and CS then reviewed the points to seek similarities and differences, to allow KP and CS to develop a list of themes. Once provisional themes were developed KP wrote up a coding framework to clearly define each theme. The development of themes was supported by use of QSR NVivo programme. The coding framework, listing the themes their codes and their descriptions, was updated throughout the analytical process, and reviewed by the wider research team, in line with qualitative rigour.

## 3. Results

### 3.1. Characteristics of the Facebook Groups

From the twelve Facebook groups, 297 posts were collected (brachycephalic = 177 posts, non-brachycephalic = 120 posts) (Table 1). Facebook groups ranged in size, from 1700 members to 51,000 members. All except one (Pug Lovers UK) were very active groups (over ten posts per day) and all were administrated and moderated by several people. The group rules within each Facebook group are presented in the Supplementary Information (Appendix A). The impact of the group rules was discussed within the research team. The rule of most relevance to our research was “This group is not to be used in place of veterinary advice. If you are concerned about your pet, we advise that you contact your veterinary practice/out of hours service.” However, although this rule was stated for five of the six non-brachycephalic groups, and one of the six brachycephalic groups, there appeared to be little, if any, impact upon the health-related content.

Following thematic analysis of the 297 posts, three overarching themes were identified: advice seeking (78%), advice giving (6%) and community bonding mechanisms (16%), described alongside their related sub-themes in Table 2. These themes will be discussed below in relation to each research objective.

### 3.2. Objective 1: Compare the Purpose of Posts in Relation to Common Health Issues between Brachycephalic and Non-Brachycephalic Owners in Breed Specific FB Groups

The purpose of the included posts varied between brachycephalic and non-brachycephalic breed groups. A graphical illustration of this is shown in Figure 2.

Advice seeking represented three quarters of the collected FB posts and highlights the importance of collective knowledge-sharing and mutual support as members addressed a wide spectrum of enquiries together. Themes such as pre-purchase research, dog ownership experiences, preventative health interventions and decision-making on corrective surgery were identified within advice seeking posts. Posts within brachycephalic breed group more frequently sought information regarding pre-purchase research and decision-making support in relation to corrective surgery, than posts identified in the non-brachycephalic breed groups.

Across both breed groups, advice-giving post themes centred around raising awareness and sharing knowledge, demonstrating the informative and collaborative nature of these FB groups. Raising awareness posts involved the sharing of external sources of information which promoted better breeding or material aimed at dissuading purchase of the breed. Within the sharing knowledge theme, discussions focused on owners experiences of symptoms and health testing. The relatively small number of quotes under this theme prevents further analysis between brachycephalic and non-brachycephalic breed owners.

Community bonding themed posts represented around one fifth of all included posts. Themes related to seeking social support and sharing experiences emerged within community bonding mechanisms, highlighting the intentional and organic strategies that contribute to building a supportive and successful community of dog breed owners. Brachycephalic breed groups contributed significantly more posts under this theme than the non-brachycephalic groups. These findings illuminate the distinct patterns of engagement and interaction within breed-specific FB communities, offering valuable insights into the dynamics of health-related discussions and the different FB behaviour of brachycephalic and non-brachycephalic breed owners.

### 3.3. Objective 2: Explore Owner Awareness of Health Issues Affecting Their Breed and Identify If This Varies between Brachycephalic and Non-Brachycephalic Dog Owners

An early distinction between brachycephalic and non-brachycephalic groups emerges in the pre-purchase research theme, within advice seeking. This theme encapsulates discussions dedicated to gathering information prior to dog ownership and includes three sub-themes: common health issues, ideal parent hip scores and concerns related to bad breeding.

Questions around common health issues indicated the potential awareness of breed-specific health concerns among prospective owners. Non-brachycephalic breed enquirers tended to exhibit a greater level of awareness of breed-specific health issues, asking specific condition-related questions such as the commonality of hip dysplasia and ideal hip scores. In contrast, individuals considering brachycephalic breeds appeared more cognisant of the breed’s potential health issues. They refrained from delving into the issue specifics or asking direct questions, taking a more generalized, or vague, approach when enquiring about health. For example, enquirers mentioned knowing that these breeds frequently visit the vet yet made no further indication as to why.

Precautions taken by owners to prevent hip dysplasia development or exacerbation were commonly demonstrated within non-brachycephalic breed groups. There was heightened awareness of health issues and proactive efforts to mitigate potential health concerns in their dogs. For example, discussing alternative and complementary therapies, activity level and genetics. Non-brachycephalic owners commonly discussed the support they were offering their dogs before considering any surgical treatment, predominant in the preventative health intervention theme. For example, pain management, alternative therapies, dietary adjustments and training. This level of practical care was noticeably absent among brachycephalic posts.

Within advice giving, the disparities in health awareness between the two groups became evident. Non-brachycephalic owners demonstrated familiarity with treatment options, which was not apparent in brachycephalic groups. Brachycephalic groups were more passive advice givers, sharing infographics and articles containing guidance on health testing, along with posts from veterinary clinics aimed at enhancing awareness of BOAS and the identification of symptoms. These posts were mainly shared from organizations such as the Cambridge BOAS Research Group, RSPCA and the Bulldog Breed Council, perhaps suggesting that some brachycephalic breed owners follow these organisations on FB. There was evidence of external influencers actively raising awareness about breed-specific health issues to educate and inform group members, yet this was minimal and surprisingly received negative responses from the group.

Commonalities in owner awareness between the two groups were demonstrated within two sub-themes of the dog ownership experiences category. Firstly, owners enquire about potential symptoms of BOAS or hip dysplasia, indicating their awareness of these conditions and their ability to recognize certain symptoms in their dogs. Secondly, both groups sought information about the long-term management of breed-specific health conditions in young dogs. This included queries related to exercise, non-surgical treatment options and their dogs’ overall quality of life. Typically, these questions arose after an owner received a veterinary diagnosis of the health condition, indicating that owners were actively taking steps to gather additional information to assist in understanding. Notably, discussions regarding the long-term management of senior dogs were exclusively observed within the non-brachycephalic groups. Finally, concerning owner awareness of common health issues, similarities between the groups emerged. Members across groups frequently shared videos of their dogs and solicited opinions from others regarding veterinary clinician advice. Frequently the topic of the requirement for, or benefits of, second opinions arose, and they were often accompanied with concerns about how to finance these and insurance coverage.

### 3.4. Objective 3: Explore the Role of Social Support Sought and Provided by Dog Owners within FB Breed Groups and Identify If This Varies between Brachycephalic and Non-Brachycephalic Dog Owners

Whilst both breed groups engaged in seeking and providing social support, it appeared most prevalent within the brachycephalic breed groups. Seeking social support and sharing of experiences within these groups appeared to be used as a mechanism for community bonding. A common example was the seeking of emotional support or soliciting good luck when sharing a personal experience.

It was only within the brachycephalic groups where the normalisation of health symptoms was observed. Normalising health symptoms was evidenced through posts seeking reassurance from the group that what their dogs were experiencing is standard for the breed. This was also evidenced in the reactions to negative media attention about their breed, where group members come together to share their dislike and anger, feeling a sense of injustice about the reporting of poor health when the issues presented were ‘normal’ for the breed. Many members took the media attention personally, commenting that they are not ‘evil’ or ‘abusive’ for owning a brachycephalic breed.

Using online breed groups as a platform for social engagement was an apparent strategy used by both brachycephalic and the non-brachycephalic owners. Group members willingly contributed to fostering a positive community by sharing their own experiences and knowledge, with the aim of empowering fellow members to make well-informed decisions. This included follow-up advice to members who had commented on their post, takin the time to communicate and increase online community bonds. The act of sharing experiences emphasises the significance of personal narratives in building connections and providing support. Both groups engaged in this practice by sharing their veterinary or surgical experiences, both positive and negative and by providing periodic updates, the majority including photos and thanking the group for their support.

## 4. Discussion

To the best of our knowledge, this is the first study which has explored and compared the content of social media posts in different dog breed groups in relation to how health information is used and appraised. There appeared to be little difference in the purpose of FB posts between breed groups suggesting that online interactions regarding breed health issues are similar across both breed owner groups. The study found that in relation to health discussions, owners used these groups to seek advice, give advice and to develop social connections with other owners. Brachycephalic breed owners were especially active in community-bonding activities, through seeking and providing social support. Sharing personal experiences played a vital role in building connections and providing support across both breed groups. Non-brachycephalic owners appeared to exhibit greater awareness of breed-specific health symptoms and long-term management of conditions.

Owners, regardless of their dog’s breed type, were generally aware of breed-specific health issues. Social media platforms serve as essential tools for information dissemination and awareness-raising. Owners often enquired about health issues prior to acquiring dogs, indicating a proactive approach to pet ownership within these groups. Non-brachycephalic dog owners seemed to exhibit a more specific awareness of potential health issues, characterised by questions regarding breed-specific conditions. In contrast, prospective brachycephalic breed owners, whilst generally aware of susceptibility to health issues, tended to adopt a less specific approach to health enquiries. It has been reported elsewhere [3,13] that brachycephalic owners prioritise appearance and behavioural attributes over health when choosing their breed, and this may explain the apparent lack of specific awareness about potential health issues. It has also been shown that owners of brachycephalic breeds view a number of symptoms of poor health as being ‘normal for the breed’ [3]; they are aware of the symptoms, but do not make a direct link with poor health, instead believing them to be an attribute distinct to their chosen breed. This might suggest that the desire to own a particular breed is more likely to override health concerns within this demographic.

There is a symbiotic relationship between owners’ health awareness and social support. Within the FB groups, owners can exchange information, experiences and advice related to their dogs’ health. This sharing of insights can enhance owner awareness of breed-specific health issues. When owners feel supported by their FB community, they are potentially more likely to seek out and engage with credible resources to proactively address health concerns [16,17]. Furthermore, shared experiences and social connections can foster a sense of responsibility and commitment to the wellbeing of their dogs, motivating owners to stay informed about potential health issues. Social support acts as a conduit for the dissemination of knowledge [17] and this could lead to more informed and health-conscious dog owners.

One significant implication of this study is that veterinarians may need to tailor communication and recommendations to meet the needs of different breed-specific owning clients. Increased owner awareness can lead to preventative care and early detection of health issues, potentially reducing the severity and cost of treatment. Veterinary clinics could further enhance their involvement in breed-specific FB groups by actively sharing resources, addressing common enquiries and raising awareness about breed-specific health issues. Whilst they can only provide specific advice to registered patients, maintaining an active veterinary presence within groups can help to foster a sense of assurance among owners, strengthening the overall support network and understanding of health issues in a non-judgmental way. The messaging and format of educational interventions to be delivered by veterinarians and animal welfare representatives warrants further research to ensure a move away from paternalistic approaches which could deter information seeking behaviours from some owners.

Our findings show that owners actively share information and offer genuine advice and support within their community, contributing to a collective understanding of breed-specific health issues and potential treatment options. These FB groups have demonstrated that many owners are actively concerned about their dogs’ wellbeing, and this might offer potential opportunities to improve canine welfare. Welfare organisations could use this insight to shape their campaigns by advocating for a more positive and constructive approach towards breed related issues.

Thematic analysis, while valuable for exploring qualitative data, is open to bias from subjectivity. To counter this, we developed a clear coding framework describing data allocated to each coded theme and agreed this framework between the team members. We recognize the potential sample bias arising from using breed-specific groups, as not all dog owners are members of such groups, and they may attract individuals with a particular level of interest or expertise. However, as noted above, this demographic appears to have been largely ignored in the scientific literature to date. Accordingly, we acknowledge that our findings may not be representative of the broader population of dog owners. A particular challenge to this type of work is the lack of opportunity to clarify understanding when analysing content extracted from social media. A related factor which may limit how representative this study is our inclusion of only FB groups and not groups on other social media platforms. Another factor influencing data available is the rules of the FB group (Appendix A). This influenced what group members could and could not post. For instance, non-brachycephalic group rules often included the need to seek veterinary advice before posting in the FB group about health issues, a practice not as prevalent in brachycephalic groups. Consequently, non-brachycephalic owners would be more inclined to share their experiences after receiving a diagnosis, enhancing their awareness and knowledge of the health issues in question. The adoption of this practice by all groups might therefore enhance the quality of any health discussion. As stated, several limitations in relation to our sample may limit how generalizable these findings are beyond the groups observed in the present study. Additional research is required to corroborate (or refute) our findings, in other dog breed groups across the spectrum of social media modalities.

Having identified the nature of content, future research could assess the quality of information and extent to which enquiries are adequately addressed from a scientific perspective. An exploration of the motivations behind pet owners seeking health advice through social media platforms would also be useful. It would then be possible to ascertain whether increased knowledge and awareness translates into improved health practices and outcomes for specific breeds. Additionally, research exploring the long-term effects of online support and information sharing on breed-specific health trends and veterinary visit patterns, would provide a more comprehensive understanding of the impact of online communities on canine health.

## 5. Conclusions

This study identified three main purposes behind discussing health issues in FB posts: to seek advice, give advice and to build social connections with other owners. Both brachycephalic and non-brachycephalic breed groups were aware of health concerns relevant to their breed, the non-brachycephalic owners appeared to display greater awareness of breed-specific health issues, whereas the brachycephalic groups appeared to put greater emphasis on providing social support for the issues raised. The use of social media groups to solicit health information highlights the need for the accurate dissemination of health and breeding information across all dog breed media channels, and the potential importance of engagement by suitable authorities in a way that is acceptable to the community. Those involved in promoting pet health and owner education should consider how their owners may be utilizing social media groups and be aware of competing information sources which may not always support the message they are trying to disseminate.

## Figures and Tables

**Figure 1 animals-14-00757-f001:**
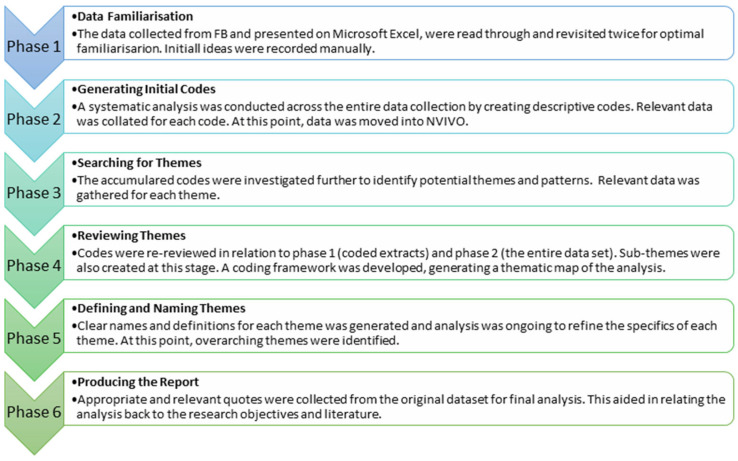
Details the 6-phase thematic analysis approach whilst visually representing each stage in the data analysis process.

**Figure 2 animals-14-00757-f002:**
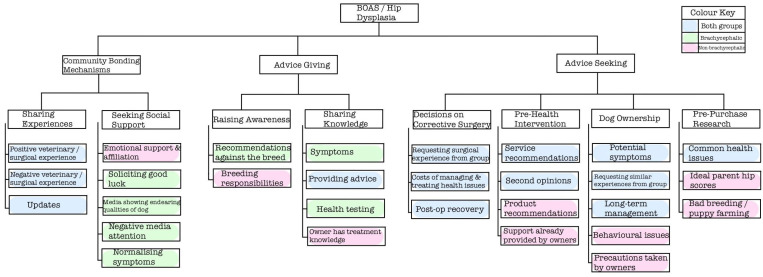
Thematic tree showcasing the three overarching themes, eight themes and the corresponding sub-themes, according to the type of group in which they were discussed.

**Table 1 animals-14-00757-t001:** Facebook group descriptors and number of posts collected per group.

Facebook Group Name	Member Count	Activity Level	Admin & Moderators	Posts Analysed
**Brachycephalic Breeds**				
**Pug Lovers UK**	4.7 K	10 posts per week	3 admin, 1 moderator	30 posts
**Pug Lovers UK** **™**	10.1 K	10 posts per day	4 admin, 2 moderators	34 posts
**French Bulldogs in the UK**	51 K	10+ posts per day	13 admin	27 posts
**French Bulldogs UK**	47 K	10+ posts per day	4 admin, 12 moderators	43 posts
**English Bulldogs UK ONLY**	15.8 K	10+ posts per day	3 admin	35 posts
**English Bulldogs UK**	1.7 K	3 posts per day	3 admin, 1 moderator	8 posts
**Non-Brachycephalic Breeds**				
**Labradors UK**	47 K	10+ posts per day	13 admin, 6 moderators	20 posts
**Labrador Owners UK**	30.9 K	10+ posts per day	2 admin, 4 moderators	20 posts
**Golden Retrievers UK**	13 K	10+ posts per day	4 admin	20 posts
**Golden Retriever Owners UK**	13 K	10+ posts per day	3 admin, 1 moderator	20 posts
**German Shepherd Family UK**	11.5 K	10+ posts per day	11 admin	20 posts
**German Shepherd UK**	15.6 K	10+ posts per day	12 admin	20 posts

**Table 2 animals-14-00757-t002:** Coding framework presenting the themes, descriptions and prevalence between brachycephalic and non-brachycephalic breed groups.

Overarching Theme	Sub-Theme	Description	Brachycephalic Frequency	Non-Brachycephalic Frequency
**Advice Seeking**		FB users actively seek guidance, insights or recommendations from the collective knowledge and experiences of group members. It reflects the fundamental value of online communities in facilitating the exchange of information and mutual support among users.	*n =* 123	*n* = 109
	Pre-purchase research (3%)	Group members engage in conversations, inquiries and information-sharing activities related to researching and preparing or the acquisition of a new dog. It can reflect a conscientious and responsible approach of individuals pre-acquisition. It may include breed-specific health enquiries, ethical and responsible ownership, breeder and adoption guidance and financial considerations.	*n* = 3	*n* = 4
	Dog ownership experiences (35%)	The day-to-day journey of dog ownership and the various challenges it presents. Advice is sought to enhance their understanding and management of life with their dog. It may include potential ill-health signs, behavioural guidance, practical tips and recommendations.	*n* = 43	*n* = 39
	Preventative health intervention (33%)	Focusing on proactive steps, preventative measures and strategies aimed at safeguarding and enhancing the overall health and wellbeing of the dog. Groups members actively seek advice and share knowledge to reduce the risk of health issues. It may include service and product recommendations and requesting second opinions.	*n* = 15	*n* = 61
	Decisions on corrective surgery(29%)	Seeking advice related to difficult and complex decisions that dog owners must take when considering corrective surgery to resolve health issues. They request personal experiences and advice potentially to make better informed choices. It may include treatment options, risks and benefits, financial considerations, quality of life, shared experiences and post-operative care.	*n* = 62	*n* = 5
**Advice Giving**		Encapsulating the act of group members sharing a variety of content, including first-hand experiences, stories, advice, resources, photos and more. It represents the communal spirit and social support that is found within FB groups.	*n* = 11	*n* = 6
	Sharing Knowledge (65%)	Active participation of group members offering knowledge, experience and insights to assist fellow members. It highlights the supportive and collaborative nature of FB groups. It may include highlighting health symptoms, providing knowledge of treatment options and veterinary professionals offering their expertise.	*n* = 6	*n* = 5
	Raising awareness (35%)	Pro-active efforts of group members to disseminate knowledge, promote education and advocacy for positive change. It may include support for animal welfare, breeding ethics and awareness of negative services.	*n* = 5	*n* = 1
**Community Bonding Mechanisms**		Various strategies and elements that contribute to building and strengthening the sense of community and shared identity within the group. It reflects the importance of a welcoming, empathetic and cohesive environment where members connect, share and support each other.	*n* = 43	*n* = 5
	Seeking social support (33%)	Group members actively seek emotional support, guidance and affiliation. It features the role of the group as a safe source of understanding and social support for members who may be dealing with various challenges. It may include open sharing of concerns, empathetic responses, validation seeking and positive reinforcement.	*n* = 14	*n* = 2
	Sharing experiences (67%)	Group members candidly share their personal stories as dog owners. It highlights the role of shared experiences in building an understanding online community. The value of personal narratives in building connections and providing support is emphasised. It may include personal anecdotes, challenges and triumphs and any updates to previous posts.	*n* = 29	*n* = 3

## Data Availability

The data presented in this study is contained in the article.

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
