# Peer review of "“Have You Seen This Drivel?” A Comparison of How Common Health Issues Are Discussed within Brachycephalic and Non-Brachycephalic Dog Breed Groups on Social Media"

_animals, 2024, doi:10.3390/ani14050757_

Round 1

Reviewer 1 Report

Comments and Suggestions for Authors

Introduction (Lines 153-205): The introduction effectively sets the context by highlighting the rising popularity of brachycephalic breeds and associated health concerns. However, it could benefit from a more direct statement of the research question and objectives. Consider revising to make the research aims more explicit early in the section.

Materials and Methods (Lines 264-322): The methodology is clear and detailed, especially in the selection of dog breeds and health conditions (Lines 276-283). However, the section on data analysis (Lines 322-325) could be expanded to provide more insight into the qualitative analysis process.

The selection of breeds and health conditions is well-explained (Lines 276-283). However, a deeper rationale for the choice of keywords for data extraction (Lines 294-302) would enhance the methodological clarity.

The data extraction process (Lines 306-312) could benefit from additional details on how the posts were stored and organized for analysis. Clarifying this could strengthen the reproducibility of the study.

Results (Lines 418-837): The results are comprehensive and well-organized, particularly the thematic analysis in Tables 1 and 2. However, the section could be enhanced by including more direct comparisons between the brachycephalic and non-brachycephalic groups, particularly in how these groups seek and share information.

While the themes identified in the posts are well-presented, it might be useful to provide more direct examples or quotes to illustrate these themes (Lines 552-555). This would add depth and clarity to the findings.

The distinction between brachycephalic and non-brachycephalic groups in terms of the content and focus of their posts (Lines 772-857) is insightful. Expanding on how these differences impact owner behavior or awareness could be valuable.

Discussion (Lines 1054-1103): This section effectively interprets the results, especially in relation to the role of social media in pet health information (Lines 1054-1065). However, the discussion could be strengthened by addressing potential limitations of the study, such as the generalizability of findings beyond the specific social media platforms studied.

The discussion effectively ties the findings to broader implications for pet health communication (Lines 1055-1065). However, exploring the potential impact of the study's findings on veterinary practice and pet owner education in more depth would be beneficial.

Addressing limitations such as potential bias in the selection of Facebook groups and the generalizability of the findings (Lines 1169-1186) is good. Further elaboration on how these limitations might impact the study's conclusions would strengthen this section.

Conclusion (Lines 1238-1248): The conclusions succinctly summarize the main findings. Consider adding suggestions for future research or implications for practice to provide a more rounded ending to the paper.

Author Response

We would like to thank the reviewer for their careful review of our manuscript. We will address each of the points you have raised below.

  1. Introduction (Lines 153-205):The introduction effectively sets the context by highlighting the rising popularity of brachycephalic breeds and associated health concerns. However, it could benefit from a more direct statement of the research question and objectives. Consider revising to make the research aims more explicit early in the section.

We thank the reviewer for this helpful comment and we have now added our study aim and objectives to the end of the introduction (lines 91-97)

  1. Materials and Methods (Lines 264-322):The methodology is clear and detailed, especially in the selection of dog breeds and health conditions (Lines 276-283). However, the section on data analysis (Lines 322-325) could be expanded to provide more insight into the qualitative analysis process.

We have added some additional text in this section. “The 6-Phase Braun and Clarke [30] approach to thematic analysis was adopted, de-scribed in Figure 1.  This approach involves inductive (data driventhemes are developed based upon similar issues or points within the data) and deductive methods (applying existing concepts and ideas from previous work) for developing themes, useful units of data to be used to demonstrate and explain the findings. KP and CS reviewed the data (comments extracted from the FB groups) and noted the points being made in each post. KP and CS then reviewed the points to seek similarities and differences, to allow KP and CS to develop a list of themes. Once provisional themes were developed KP wrote up a coding framework to clearly define each theme. The development of themes was supported by use of QSR NVivo programme. The coding framework, listing the themes their codes and their descriptions, was developed and updated throughout the analytical process, and reviewed by the wider research team, in line with qualitative rigour.” (lines 171-178)

  1. The selection of breeds and health conditions is well-explained (Lines 276-283). However, a deeper rationale for the choice of keywords for data extraction (Lines 294-302) would enhance the methodological clarity.

Thank you for raising this. We have amended this section to read: “Key words to identify the first 20 relevant posts in each group were determined through a collaborative decision-making process within the research team, which included experts in veterinary medicine, qualitative research, marketing and social media research. Key words were selected to include medically correct terminology and lay terminology known to be used by owners when discussing these health conditions. Sense checking in the Facebook groups was conducted to ensure they were appropriate and providing relevant results (lines 134-137)

  1. The data extraction process (Lines 306-312) could benefit from additional details on how the posts were stored and organized for analysis. Clarifying this could strengthen the reproducibility of the study.

We are unclear how much more detail we can add here, but we have confirmed in the manuscript that the posts were copied and pasted into the Excel sheet. We do not used any data extraction software for this task. (lines 147-148)

  1. Results (Lines 418-837):The results are comprehensive and well-organized, particularly the thematic analysis in Tables 1 and 2. However, the section could be enhanced by including more direct comparisons between the brachycephalic and non-brachycephalic groups, particularly in how these groups seek and share information.

Thank you for this suggestion. We are unable to present direct quotes as those who we observed in the groups have not provided informed consent. This also to some degree restricts use of examples where care must be taken when we give examples that the individual remains anonymous. We have however expanded several points throughout the results section that we hope provides the reader with more depth and clarity.

Questions around common health issues indicated the potential awareness of breed-specific health concerns among prospective owners. Non-brachycephalic breed enquirers tended to exhibit a greater level of awareness of breed-specific health issues, asking specific condition-related questions such as the commonality of hip dysplasia and ideal hip scores. In contrast, individuals considering brachycephalic breeds appeared more cognisant of the breed’s potential health issues. They refrained from delving into the issue specifics or asking direct questions, taking a more generalized, or vague, approach when enquiring about health. For example, enquirers mentioned knowing that these breeds frequently visit the vet yet made no further indication as to why. (lines 256-264)

Within advice giving, the disparities in health awareness between the two groups became evident. Non-brachycephalic owners demonstrated familiarity with treatment options, which was not apparent in brachycephalic groups. Brachycephalic groups were more passive advice givers, sharing infographics and articles containing guidance on health testing, along with posts from veterinary clinics aimed at enhancing awareness of BOAS and identification of symptoms. These posts were mainly shared from organizations such as the Cambridge BOAS Research Group, RSPCA, and the Bulldog Breed Council, perhaps suggesting that some brachycephalic breed owners follow these organizations on FB There was evidence of external influencers actively raising awareness about breed-specific health issues to educate and inform group members, yet this was minimal and surprisingly received negative responses from the group.   (lines 277-287)

It was only within the brachycephalic groups where the normalisation of health symptoms was observed. Normalising health symptoms was evidenced through posts seeking reassurance from the group that what their dogs were experiencing is standard for the breed. This was also evidenced in the reactions to negative media attention about their breed, where group members come together to share their dislike and anger, feeling a sense of injustice about the reporting of poor health when the issues presented were ‘normal’ for the breed. Many members took the media attention personally, commenting that they are not ‘evil’ or ‘abusive’ for owning a brachycephalic breed. (Lines 318-325)

Using online breed groups as a platform for social engagement was an apparent strategy used by both brachycephalic and the non-brachycephalic owners. Group members willingly contributed to fostering a positive community by sharing their own experiences and knowledge, with the aim of empowering fellow members to make well-informed decisions. This included follow-up advice to members who had commented on their post, takin the time to communicate and increase online community bonds. The act of sharing experiences emphasises the significance of personal narratives in building connections and providing support. Both groups engaged in this practice by sharing their veterinary or surgical experiences, both positive and negative and by providing periodic updates, the majority including photos and thanking the group for their support. (lines 328-337)

  1. While the themes identified in the posts are well-presented, it might be useful to provide more direct examples or quotes to illustrate these themes (Lines 552-555). This would add depth and clarity to the findings.

Thank you for this suggestion. We have expanded several points throughout the results section that we hope provides the reader with more depth and clarity as detailed under our response for point 5.

  1. The distinction between brachycephalic and non-brachycephalic groups in terms of the content and focus of their posts (Lines 772-857) is insightful. Expanding on how these differences impact owner behavior or awareness could be valuable.

Thank you for this suggestion. The line numbers given are not corresponding with our paper but as indicated above we have expanded several points in the results section and we hope this adds some clarity and further refinement to our findings.

  1. Discussion (Lines 1054-1103):This section effectively interprets the results, especially in relation to the role of social media in pet health information (Lines 1054-1065). However, the discussion could be strengthened by addressing potential limitations of the study, such as the generalizability of findings beyond the specific social media platforms studied.

Thank you. We have added “A related factor which may limit how representative this study is our inclusion of only FB groups and not groups on other social media platforms”. (line 411-413)

  1. The discussion effectively ties the findings to broader implications for pet health communication (Lines 1055-1065). However, exploring the potential impact of the study's findings on veterinary practice and pet owner education in more depth would be beneficial.

Thank you for this comment. We think addressing this in detail may be beyond the scope of our study and its findings. There is certainly the potential to impact on practice and owner education as has been stated but further work is required to fully understand this.

  1. Addressing limitations such as potential bias in the selection of Facebook groups and the generalizability of the findings (Lines 1169-1186) is good. Further elaboration on how these limitations might impact the study's conclusions would strengthen this section.

Thank you. We have added “As stated, several limitations in relation to our sample may limit how generalizable these findings are beyond the groups observed in the present study. Additional research is required to corroborate (or refute) our findings, in other dog breed groups across the spectrum of social media modalities.” (lines 420-424)

  1. Conclusion (Lines 1238-1248):The conclusions succinctly summarize the main findings. Consider adding suggestions for future research or implications for practice to provide a more rounded ending to the paper.

Thank you. We have added “. Those involved in promoting pet health and owner education should consider how their owners may be utilizing social media groups and be aware of competing information sources which may not always support the message they are trying to disseminate”. (line 445-447)

Reviewer 2 Report

Comments and Suggestions for Authors

Dear authors,

I have read your study, and while I think there is a potential to it and its results, there are some major issue I found with the methodological approach as well as the presentation of results and subsequent discussion. Moreover, there are some major flaws to writing itself and very often the text is very chaotic and some thoughts are repeating throughout the results and discussion in a different wording. Below you will find comments to some of the sections:

Methodology

Overall, the study is poorly designed and does not include a lot of methodological work, which could provide solid results. 

1. Considering the little work necessary for gathering results, I do not understand why did you include only 6 different breeds (Lin 115) and identified only 20 relevant posts (Lin 121) from each social group. This does not provide enough data that could be statistically significant.

2. In relation to the previous comments, there is a complete lack of any statistical analysis to show the significance of your observations. The way they are presented is purely subjective.

3. You say (Lin 135) that you included only posts with 5 and more comments. Have you taken into consideration, that most of the posts, would have also “passive observers” (people who would base their actions based on the post without providing any feedback such as comment or “like”). Why didn’t you take into consideration the total amount of views of the different posts as well?

4. Lin 140. Do you mean that the ethical committee accepted your design completely, or there were some reservations?

5. Lin. 159. There is no need to repeat the reference in the Figure description, as long as its mentioned in the text.

6. Why didn’t you include a questionnary as another form of data collection? It would provide a comparable dataset for the open form that is already presented.

Results

The text is very chaotic and the sectioning based on “objectives” resembles that of some thesis rather than a peer-reviewed article. Moreover, there is no clear distinction between some parts of it, and that what should a reader expect in the Discussion.

1. Lin. 164. Why is there such a gap in the number of posts between the two groups. This immediately leads to problem with comparability between the two data sets. Especially without any statistical analysis.

2. Lin. 166. Which was not an active group? You need to be more specific.

3. Table 1. The activity level should show the average number of posts per group. Especially if there are such vast differences between the member counts. Moreover, are the posts counted in the column “activity level” those that are relevant to the health issues or does it stand for the total amount of posts?

4. Lin. 180. It should be expected that the results will be further discussed. No need to mention that.

5. Table 2. The frequencies need to be presented not only in their total values but also in percentages so that there is a clear comparison between the sub-themes.

6. Figure 2. The figure is absolutely unreadable in its size/form.

7. Lin 196 and 214. It is important to use exact values and percentages when describing proportions. This issues appears regularly throughout the results section. Terms such as “more passive”, “around” or “greater” are more fit for the Discussion.

8. Lin 222-224. I assume you have forgotten a part of a co-author’s comments here.

Discussion

Similar to the results, it is very poorly written and lacks any structure. Moreover, there are too many speculations (e.g. LIN 326 - “seemed to exhibit”, LIN 333 – “might suggest”) in comparison to statements based on any scientific proofs.

I highly suggest that you provide a complete overhaul of this section together with some parts of the Results. There are many other issues with the wording, format or reference choices, however, I believe that I have provided enough feedback for you to improve the study on.

Author Response

We would like to thank the reviewer for their careful review of our manuscript. We will address each of their points below.

I have read your study, and while I think there is a potential to it and its results, there are some major issue I found with the methodological approach as well as the presentation of results and subsequent discussion. Moreover, there are some major flaws to writing itself and very often the text is very chaotic and some thoughts are repeating throughout the results and discussion in a different wording. Below you will find comments to some of the sections:

Methodology

Overall, the study is poorly designed and does not include a lot of methodological work, which could provide solid results. 

  1. Considering the little work necessary for gathering results, I do not understand why did you include only 6 different breeds (Lin 115) and identified only 20 relevant posts (Lin 121) from each social group. This does not provide enough data that could be statistically significant.

We thank the reviewer for this comment. Our paper presents a qualitative analysis and not quantitative, therefore statistical significance is not relevant. As presented in the paper (lines 135-136) and is standard in qualitative research, data saturation, where no new themes can be found in the data, informs us that we have collected enough data. For clarity we have made some minor amendments in the method section to highlgiht this is a qualitative content analysis. The analysis is primarily qualitiative, using thematic analysis, with a limited use of basic descriptive quantitative results to describe the frequency of themes.

  1. In relation to the previous comments, there is a complete lack of any statistical analysis to show the significance of your observations. The way they are presented is purely subjective.

We thank the reviewer for this comment.  However, as above, this is a qualitative study and as such statistical significance is not relevant or appropriate.

  1. You say (Lin 135) that you included only posts with 5 and more comments. Have you taken into consideration, that most of the posts, would have also “passive observers” (people who would base their actions based on the post without providing any feedback such as comment or “like”). Why didn’t you take into consideration the total amount of views of the different posts as well?

Our selection criteria were chosen to include the most relevant and useful pieces of data from the social media groups. As discussed in the paper, social media member groups are based on interactions. To capture the most relevant data, we needed to also capture the data that has been interacted with. Collecting the number of views is not always possible as this depends upon the admin settings of each individual group. It is therefore not possible to truly quantify passive observers, and as previously stated, our study focused on active engagers, and not passive observers.

  1. Lin 140. Do you mean that the ethical committee accepted your design completely, or there were some reservations?

We thank the reviewer for this comment. Full ethical approval was provided. The wording has been updated to read “This study has been reviewed and approved by…” (Line 139)

  1. 159. There is no need to repeat the reference in the Figure description, as long as its mentioned in the text.

Thank you, reference removed from figure description.

  1. Why didn’t you include a questionnary as another form of data collection? It would provide comparable dataset for the open form that is already presented.

The objective was to understand how people used social media groups to discuss common health problems associated with different breeds. We wished to observe how people behaved in this context. Questionnaires would not permit direct observation and would not address our research objective.

Results

The text is very chaotic and the sectioning based on “objectives” resembles that of some thesis rather than a peer-reviewed article. Moreover, there is no clear distinction between some parts of it, and that what should a reader expect in the Discussion.

  1. 164. Why is there such a gap in the number of posts between the two groups. This immediately leads to problem with comparability between the two data sets. Especially without any statistical analysis.

We thank the reviewer for this comment. As previously stated, this is a qualitative study and as such there is no statistical analysis.

  1. 166. Which was not an active group? You need to be more specific.

We thank the reviewer for this comment. We have added “(Pug Lovers UK)” in line 187 to indicate which was not an active group.

  1. Table 1. The activity level should show the average number of posts per group. Especially if there are such vast differences between the member counts. Moreover, are the posts counted in the column “activity level” those that are relevant to the health issues or does it stand for the total amount of posts?

The activity level reported in table 1 is the value descriptor given by Facebook itself. It is not specific to health posts. To the best of our knowledge this is an average determined by Facebook. It is being used as a descriptor of the activity level of the group i.e. is it a busy and frequently used group versus a quieter and less commonly used group.

  1. 180. It should be expected that the results will be further discussed. No need to mention that.

We thank the reviewer for this comment. Signposting the reader is common when presenting qualitative findings

  1. Table 2. The frequencies need to presented not only in their total values but also in percentages, so that there is a clear comparison between the sub-themes.

We thank the reviewer for this comment. For small numbers in qualitative research we would not normally use percentages as they are not overly relevant in qualitative research.

  1. Figure 2. The figure is absolutely unreadable in its size/form.

We have submitted a full A4 sized copy of the image.

  1. Lin 196 and 214. It is important to use exact values and percentage when describing proportions. This issues appears regularly throughout the results section. Terms such as “more passive”, “around” or “greater” are more fit for the Discussion.

We thank the reviewer for this comment. This is not required for qualitative research. The use of words to indicate size, range and number is preferable in qualitative research.

  1. Lin 222-224. I assume you have forgotten a part of a co-author’s comments here.

Thank you, this has been removed. This was hang over text from the template provided for the manuscript.

Discussion

  1. Similar to the results, it is very poorly written and lacks any structure. Moreover, there are too many speculations (e.g. LIN 326 - “seemed to exhibit”, LIN 333 – “might suggest”) in comparison to statements based on any scientific proofs. I highly suggest that you provide a complete overhaul of this section together with some parts of the Results. There are many other issues with the wording, format or reference choices, however, I believe that I have provided enough feedback for you to improve the study on.

We thank the reviewer for this comment. The purpose of qualitative methods is that they are exploratory and hypotheses generating. Qualitative research does not permit hypothesis testing, therefore conclusive statements would be an inappropriate output of qualitative research. The themes have been clearly presented as shown in table 2 and reported under their respective research objectives in the manuscript. We have made several minor changes in line with the opinions of the other reviewers but we do not feel this section requires significant re-writing.

Reviewer 3 Report

Comments and Suggestions for Authors

Dear Authors,

I really appreciate your tremendous effort you did in this study to submit the manuscript. By my standpoint, you worked hard but the manuscript to be accepted for publication I consider reasonable to do the corrections included in the attachment

Author Response

We would like to thank the reviewer for their careful review of our manuscript. We will address each of their points raised below.

  1. Row 11, the most appropriate term is „short-nose” instead of flat faced, please replace it

Thank you We have replaced flat-face with short-nose throughout the manuscript.

  1. Rows 16/17, „…We selected 12 breed groups, 6 for flat faced breeds and 6 for non-flat faced breed….” versus Rows 106-109 „….The condition of hip dysplasia was chosen for the non-brachycephalic groups, which was to include breed groups for Golden Retriever, Labrador and German Shepherd. The condition of Brachycephalic Obstructive Airway Syndrome(BOAS) chosen for the brachycephalic groups which was to include Bulldog, French Bulldog and Pug….” Please revise and correct the sentence in rows 16/17, accordingly with 2 Data Collection, you in reality selected just 6 breeds

We have changed to read “We selected 6 dog breeds (3 flat faced and 3 non-flat faced) and identified two breed groups for each breed” (lines 16-17) to clarify.

  1. Row 48 …..not least airflow resistance and obstructed breathing,… should add „ including the secondary gastrointestinal(GI) disorders and hiatal hernia

Added as suggested, thank you.

  1. Row 109, please insert the full name of the breed: English Bulldog, not just Bulldog

Now corrected, thank you

  1. Rows 125/126 ….Soft palette…., please correct „soft palate

Now corrected, thank you.

Reviewer 4 Report

Comments and Suggestions for Authors

Reviewer comments for manuscript ID animals-2855608 entitled “Have you seen this drivel?” A comparison of how common health issues are discussed within brachycephalic and non-brachycephalic dog breed groups on social media.

General comments

Social media affects live all over the world now. It influences decision making, creates information pool and promotes social bonding through virtual platforms. Dog breed owner groups are off shoots of so many common interest groups found on the facebook that has influenced pet lovers’ decision making regarding owning specific breeds, their needs and health issues. These groups are one prior steps to veterinary consults and might provide pet owners with more knowledge that might improve their decision making while dealing with their veterinarians.

It is a unique study on the analysis of social media groups affecting the decisions, behaviour and community bonding of two broadly different breed owner groups. I congratulate the authors for this novel work. The manuscript is very well written with almost error free writing. The gaps in literature have been nicely identified and explained. The data analysis is accurate and exclusive softwares like NVivo has been used to analyse the data. Results are very nicely described and further elaborated through tables and graphs. Discussion is precise, informative, in depth and insightful.  I would like to ask the authors that did they found queries about sales of pets, dog breeders and sellers. This information can be important for future awareness campaigns and initiatives to improve pet care and scientific breeding practices. I have very few and minor suggestions/corrections that I have indicated specifically. I recommend the publication of the manuscript.

Specific comments

Line 12: I think ‘structure’ will be a better word than ‘feature’

Line 61: Should this be ‘health related’ instead of ‘related health’? Please clarify.

Table 2: What reason do you think could explain the significantly lesser number in the brachycephalic group than the non- brachycephalic group under the ‘Preventative Health Intervention’ and the other way round under the ‘Decisions on corrective surgery’?

Line 334: Brachycephalic breed owners also think as per your results that the health issues are just normal for these breeds. Please clarify and then add.

Lines 355-66: Don’t you feel such measures if adopted by the vets/clinics can significantly reduce their consults and loss of clients? Please clarify.

Lines 377-83: Did you find the group rules adhered to by the breed owners? Please clarify.

Author Response

We would like to thank the reviewer for their careful review of our manuscript. We will address each of their points below.

Specific comments

  1. Line 12: I think ‘structure’ will be a better word than ‘feature’

Corrected, thank you

  1. Line 61: Should this be ‘health related’ instead of ‘related health’? Please clarify.

Thank you, this is now corrected.

  1. Table 2: What reason do you think could explain the significantly lesser number in the brachycephalic group than the non- brachycephalic group under the ‘Preventative Health Intervention’ and the other way round under the ‘Decisions on corrective surgery’?

We thank the reviewer for this comment. The specific Facebook group rule of not asking for advice in replacement of seeking veterinary attention (as discussed below in point 6) , which was in force in a greater number of non-brachycephalic breed groups than brachycephalic breed groups, may have influenced this finding, particularly in relation to decision making about surgeries where brachycephalic owners also used the breed groups to obtain social support. It may be that the social support needs are linked to seeking support for decision making. Additionally, as noted in our manuscript, posts in the brachycephalic breed groups appeared more vague and showed little in the way of understanding the specific health risks and conditions associated with their breed. This may suggest these owners are genuinely less aware that health problems can be prevented, therefore they are not going to seek preventative advice in the first place. The limitation of direct observation as presented in this current study is the inability to follow up on emerging differences and seek to explain them. But this is one of several issues to be considered in future research.

  1. Line 334: Brachycephalic breed owners also think as per your results that the health issues are just normal for these breeds. Please clarify and then add.

We thank the reviewer for this suggestion and have added “It has also been shown that owners of brachycephalic breeds view a number of symptoms of poor health as being ‘normal for the breed’ [3]; they are aware of symptoms but do not make a direct link with poor health, instead believing them to be an attribute distinct to their chosen breed” (lines 362-365).

  1. Lines 355-66: Don’t you feel such measures if adopted by the vets/clinics can significantly reduce their consults and loss of clients? Please clarify.

The reviewer raises a valid point, and one that we cannot answer based on this study. However, based upon our findings, we would suggest that vet-owner relationship/ communication models should consider the differences between owners in relation to their information seeking behaviours, and the reasons behind these. If we consider the doctor-patient (human) relationship, we know that traditional paternalistic approaches tend to fail. When the power is shared between expert and patient/ client, there is a stronger base to provide education in a way that does not increases barriers to receiving it. In view of this, we have added “The messaging and format of educational interventions to be delivered by veterinarians and animal welfare representatives warrants further research to ensure a move away from paternalistic approaches which could deter information seeking behaviours from some owners.” (lines 389-392) as we believe the reviewer raises a valid point.

  1. Lines 377-83: Did you find the group rules adhered to by the breed owners? Please clarify.

We thank the reviewer for this comment. This is not something that we specifically measured but our experiences observing these groups suggest that yes, the rules were generally adhered to, possibly related to having good admins moderating the groups well.

The rule we consider most pertinent to our study is “This group is not to be used in place of veterinary advice. If you are concerned about your pet, we advise that you contact your veterinary practice/out of hours service” (documented in supplementary information, s1). This rule was in place in 6 of the 12 selected groups. However as indicated in our manuscript, this rule being in place had little impact on health discussions. However, of the 6 groups with this rule in place, 5 were non-brachycephalic breed groups, and so it is possible having this rule in place was steering towards the general increase in preventative health discussions that predominated these breed groups (as discussed briefly under point 3).  

Round 2

Reviewer 2 Report

Comments and Suggestions for Authors

Dear authors,

thank you for addressing some of my comments and questions. While I do not agree with some of the answers, I do not want be an obstacle for your study.  

The last thing I would however recommend is to unify to graphical side of your tables (e.g. Italics, borders).